# Skipping Breakfast for 6 Days Delayed the Circadian Rhythm of the Body Temperature without Altering Clock Gene Expression in Human Leukocytes

**DOI:** 10.3390/nu12092797

**Published:** 2020-09-12

**Authors:** Hitomi Ogata, Masaki Horie, Momoko Kayaba, Yoshiaki Tanaka, Akira Ando, Insung Park, Simeng Zhang, Katsuhiko Yajima, Jun-ichi Shoda, Naomi Omi, Miki Kaneko, Ken Kiyono, Makoto Satoh, Kumpei Tokuyama

**Affiliations:** 1Graduate School of Integrated Arts and Sciences, Hiroshima University, 1-7-1 Kagamiyama, Higashi-Hiroshima, Hiroshima 739-8521, Japan; 2Faculty of Health and Sport Sciences, University of Tsukuba, 1-1-1 Tennodai, Tsukuba, Ibaraki 305-8574, Japan; horipico@yahoo.co.jp (M.H.); yoshi.tanaka.5212@gmail.com (Y.T.); aand2.x2@gmail.com (A.A.); shodaj@md.tsukuba.ac.jp (J.-i.S.); omi.naomi.gn@u.tsukuba.ac.jp (N.O.); 3Department of Somnology, Tokyo Medical University, 5-10-10 Yoyogi, Shibuya-ku, Tokyo 151-0053, Japan; momoko-k@tokyo-med.ac.jp; 4International Institute for Integrative Sleep Medicine (WPI-IIIS), University of Tsukuba, 1-1-1 Tennodai, Tsukuba, Ibaraki 305-8550, Japan; park.insung.ge@u.tsukuba.ac.jp (I.P.); zhang.simeng.gm@un.tsukuba.ac.jp (S.Z.); mk-satoh@outlook.com (M.S.); tokuyama.kumpei.gf@u.tsukuba.ac.jp (K.T.); 5Faculty of Pharmacy and Pharmaceutical Sciences, Josai University, 1-1 Keyakidai, Sakado, Saitama 350-0295, Japan; k-yajima@josai.ac.jp; 6Graduate School of Engineering Science, Osaka University, 1-3 Machikaneyama, Toyonaka, Osaka 560-8531, Japan; kaneko@bpe.es.osaka-u.ac.jp (M.K.); kiyono@bpe.es.osaka-u.ac.jp (K.K.)

**Keywords:** skipping breakfast, core body temperature, dim light melatonin onset (DLMO), heart rate variability, clock gene

## Abstract

Breakfast is often described as “the most important meal of the day” and human studies have revealed that post-prandial responses are dependent on meal timing, but little is known of the effects of meal timing per se on human circadian rhythms. We evaluated the effects of skipping breakfast for 6 days on core body temperature, dim light melatonin onset, heart rate variability, and clock gene expression in 10 healthy young men, with a repeated-measures design. Subjects were provided an isocaloric diet three times daily (3M) or two times daily (2M, i.e., breakfast skipping condition) over 6 days. Compared with the 3M condition, the diurnal rhythm of the core body temperature in the 2M condition was delayed by 42.0 ± 16.2 min (*p* = 0.038). On the other hand, dim light melatonin onset, heart rate variability, and clock gene expression were not affected in the 2M condition. Skipping breakfast for 6 days caused a phase delay in the core body temperature in healthy young men, even though the sleep–wake cycle remained unchanged. Chronic effects of skipping breakfast on circadian rhythms remain to be studied.

## 1. Introduction

Breakfast is often described as “the most important meal of the day” [1], and while it is believed to contribute to good health and nutrition by providing essential nutrients early in the day [2]; adolescents frequently skip breakfast [3]. Many studies have reported the health benefits associated with breakfast [4,5,6,7]. A recent metabolomics study indicated that the total metabolic response tended to be larger in the morning than in the evening [8]. 

Day-to-day physiologic activities, such as the sleep–wake cycle, feeding behavior, hormone release, and body temperature, follow a daily cycle called a circadian rhythm [9,10]. The circadian components of the body temperature rhythm and melatonin secretion are controlled by the suprachiasmatic nucleus (SCN), with body temperature peaking in the late afternoon and declining to its nadir during the sleep phase in humans [11]. In general, in human clinical research, the melatonin level is considered a gold-standard proxy for the SCN phase [12]. Little is known about the effect of skipping breakfast, however, on the amplitude and phase of the body temperature rhythm in humans, because there are no reports of continuous measurement of body temperature [13,14]. In mammals, the central ‘master’ clock located in the SCN incorporates environmental information and synchronizes clock phases in peripheral tissues, i.e., peripheral clocks [15,16]. The clock genes comprise primarily two loops: a positive feedback loop and a negative feedback loop. The positive feedback loop consists of the *circadian locomotor output kaput* (*CLOCK*) and *brain and muscle ARNT-like protein 1* (*BMAL1*) heterodimer, which mediates the transcription of tissue-specific genes. The negative feedback loop consists of the *period* (*PER*) and *cryptochrome* (*CRY*) proteins, which inhibit CLOCK:BMAL1-mediated transcription [17]. In mouse studies, skipping breakfast for 4 consecutive days decreases the amplitude of the body temperature and leads to a phase delay in *BMAL1* in the liver and fat tissue [18]. In rats, skipping breakfast for 2 weeks caused a 4-h delay in the body temperature surge and phase delays in *BMAL1*, *CLOCK*, *PER2*, and *CRY1* in the hepatic tissue [19]. Previous human studies which adopted a constant routine protocol reported that leukocytes, beard hair follicle cells, and whole blood possess an endogenous circadian clock, suggesting that *PER1–3* and *BMAL1* expression are appropriate biomarkers and that these tissues would be a useful source for evaluating biologic clock traits in individuals [20,21]. To our knowledge, the effects of skipping breakfast have never been evaluated in humans using this method.

The autonomic nervous system is also controlled by the SCN and acts as a bridge between the master clock and the peripheral clocks [22]. Frequency analysis of heart rate variability can be used to evaluate the cardiac autonomic nervous system based on electrocardiography recordings [23]. A previous study reported that the dynamic balance between the sympathetic nervous system and the parasympathetic nervous system quickly responds to environmental cues, such as fasting, to appropriately adapt energy metabolism [24]. A single incident of skipping breakfast, however, does not induce changes in sympathetic and/or parasympathetic nervous system activity [25,26].

Daily meal frequency is influenced not only by biologic factors and habit, but also by social factors such as lifestyle and/or occupation, and is thought to influence weight change and glucose tolerance. Circadian clocks influence a broad range of biologic processes, including neuronal, endocrine, metabolic, and behavioral functions [27,28]. The effects of meal timing on the human circadian system, particularly the rhythms of body temperature and expression of clock genes, are poorly understood. Therefore, in the present study, we conducted breakfast-skipping experiments for 6 days, assuming that it would have a greater effect than a single incident of skipping breakfast. Although accumulated energy expenditure and substrate oxidation over 24 h were not affected by 6 days of skipping breakfast, their time courses over 24 h were significantly altered, and body weight unexpectedly increased [29]. The purpose of the present study was to evaluate the effects of skipping breakfast for 6 days on indicators of circadian rhythms, such as the 24-h core body temperature, heart rate variability, dim light melatonin onset (DLMO), and clock gene expression in leukocytes.

## 2. Materials and Methods

### 2.1. Subjects

Ten male Japanese subjects (20–30 years of age) participated in the present study. Exclusion criteria were as follows: (1) presence of food allergies, (2) occasional or habitual breakfast skippers, (3) smoking habit, (4) history of chronic disease, (5) regular use of medications, and (6) evening type, judged from a Japanese version of the morningness–eveningness questionnaire [30]. The sample size, based on the results of a preliminary study, was set at 8 per group to achieve a significance level of 5% with a detection power of 0.8. To allow for discontinuation or measurement errors during the meal intervention, 10 subjects were recruited. Power analysis was conducted using G-Power 3.1.7 software (written by F. Faul, University of Kiel, Germany). The trial was conducted from July 2015 to October 2016. This study was approved by the Local Ethics Committee of the University of Tsukuba, conducted in accordance with the principles set forth in the Declaration of Helsinki, and complied with the study protocol. All subjects provided written informed consent to participate in the study after receiving an explanation of the experiments and associated risks. The trial was registered with the University Hospital Medical Information Network (UMIN; http://www.umin.ac.jp/; Registration No. UMIN000032346).

The energy metabolism, blood glucose, and body composition measurement results from this experiment were published in another paper [29].

### 2.2. Study Protocol

This study had a cross-over design with alternating assignment. The intervention involved providing subjects with a 3-meal per day diet (3M) or a 2-meal per day diet (2M, i.e., breakfast-skipping condition) over the course of 6 days, with the conditions switched after a minimum wash-out period of 1 week. Figure 1 shows a schematic summarizing the study design. From 1 week before the meal intervention, i.e., the baseline or wash-out period, subjects were asked to maintain a regular sleep (23:00)/wake (06:00) schedule and meal schedule (07:00, 12:30, and 18:00), eating meals they each prepared for themselves. During the intervention period (Day 1 to Day 6), the subjects were instructed to eat only the meals provided by the study coordinator, which were individually adjusted according to each subject’s estimated energy requirement [31]. In each meal intervention trial, the subjects ate breakfast (33.3% of daily energy intake for the 3M condition) or no breakfast (0 kcal for the 2M condition), lunch (33.3% or 50.0% of daily energy intake for the 3M and 2M conditions, respectively), and dinner (33.3% or 50.0% of daily energy intake for the 3M and 2M conditions, respectively), such that the 24-h energy intake was equal for both dietary conditions (2197 ± 405 kcal/d, 15% protein, 25% fat, and 60% carbohydrates). The subjects were also instructed to refrain from consuming beverages containing stimulants, caffeine, or alcohol, and from performing extreme exercise during the meal intervention.

At 22:00 on the fifth meal intervention day, an intravenous catheter was placed in a forearm vein for blood sampling, and the subjects entered into a room-sized respiratory chamber where they remained for 33 h (until 07:00 of the seventh day). In the chamber, the subjects slept for 7 h from 23:00 to 06:00 the next morning and were asked to remain seated as much as possible thereafter. On the sixth-day meal intervention, the subjects were seated in dim light (<10 lux) starting 4 h before their bedtime (from 19:00 to 23:00) to assess DLMO. Under the dim light conditions, they were allowed to listen to music or read books. 

Figure 1 provides a schematic overview of the study protocol (upper panel) and time schedule of meal intervention following the baseline and/or wash-out period (lower panel). All subjects ate meals they each prepared themselves during the baseline and/or wash-out (
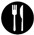
), and they ate 3 meals/day. During the meal intervention days, the subjects ate meals provided by the study (
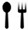
). On the fifth day of the meal intervention, the subjects arrived at the laboratory, entered the metabolic chamber at 22:00, and stayed until the morning of the seventh day (

). Before entering the metabolic chamber, each subject ingested a pill-sized CorTemp sensor, had a catheter placed in a forearm vein, and wore a telemetric heart rate monitor. From the sixth to seventh day of meal intervention, blood samples were collected at 06:00, 9:00, 12:00, 15:00, 18:00, 21:00, 23:00, and the next morning 06:00 (
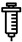
). From 19:00 to 23:00 on the sixth day of the meal intervention, each subject was seated in the dim light (<10 lux) and saliva was collected every 30 min (
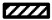
). Note that the timings of meals and sleep were predetermined by the experimental protocol

### 2.3. Measurements

#### 2.3.1. Core Body Temperature

Each subject ingested a pill-sized CorTemp sensor (HQ, Inc., Palmetto, FL, USA) before entering the chamber. The sensor measures the core body temperature every 30 s while passing through the gastrointestinal tract by delivering a radio signal to an external recorder. The CorTemp sensors were calibrated against a digital thermometer prior to being ingested. Median temperature over every 5 min was calculated, and the data were averaged per 30 min for each subject. 

#### 2.3.2. Dim Light Melatonin Onset

Salivary samples were collected using a saliva collection aid (Salivette, Sarstedt AG & Co., Nümbrecht, Germany) every 30 min from 19:30 to 23:00 on the sixth day of the meal intervention. Saliva was collected based on the standard procedures for DLMO measurements: (1) 15 min before every saliva collection, the subjects rinsed their mouths with water, (2) the subjects tipped the cotton swab from the Salivette into their mouths, and rolled the cotton swab in their mouths for 1 min until saturated, and (3) toothpaste or mouthwash were not allowed during the phase assessments [32]. These samples were immediately centrifuged to extract the saliva from the cotton swab and then frozen (−20 °C) until analysis. Saliva samples from each individual were assayed in the same batch. Melatonin levels were analyzed using a commercial enzyme-linked immunosorbent assay kit (Salimetrics LLC, State College, PA, USA) and spectrophotometer (iMark Microplate Reader, Bio-Rad, Hercules, CA, USA). 

The timing of the DLMO phase was determined by linear interpolation between the time-points before and after the melatonin concentration increased and remained above the 4-pg/mL threshold [33].

#### 2.3.3. Heart Rate and Heart Rate Variability

Each subject wore a telemetric heart rate monitor (DS-2151, Fukuda Denshi Co., Ltd., Tokyo, Japan) during the indirect calorimetry measurement. The R-R intervals of the electrocardiogram were continuously monitored, and the power spectrum of heart rate variability was estimated using the maximum entropy method. Heart rate variability reflects the balance between the activities of the sympathetic and parasympathetic components of the autonomic nervous system [23]. The spectral measures indicating contributions of variance were computed as areas under the power spectrum in subranges and are presented in square milliseconds (ms^2^). Parasympathetic and sympathetic nervous system activity were estimated as high frequency (HF; 0.15–0.4 Hz) and as the power ratio of low frequency (LF; 0.04–0.15 Hz) to high frequency (LF/HF), respectively [34]. Because there were large individual differences in HF, HF was normalized by taking the logarithm (log HF). The obtained data (heart rate, LF/HF, and log HF) were averaged for each 30-min period for each subject. 

#### 2.3.4. Clock Genes

Blood samples (2.5 mL) were collected via the intravenous catheter at 06:00, 09:00, 12:00, 15:00, 18:00, 21:00, and 23:00 on the sixth day of the meal intervention, and in the following morning at 06:00 into PAXgene Blood RNA vacutainer tubes (PreAnalytiX, Hombrechtikon, Switzerland). After mixing by inversion and incubating at room temperature for several hours, the samples were kept frozen at −20 °C until RNA extraction using the PAXgene Blood RNA kit (QIAGEN, Hilden, Germany). Extracted total RNA was analyzed spectrophotometrically (NanoVue plus, GE Healthcare Life Sciences, UK) to check the quality and quantity. Equal amounts of RNA were reverse-transcribed using a PrimeScript RT reagent Kit (Takara, Japan) and a Thermal Cycler (Takara PCR Thermal Cycler Dice Gradient, Japan) according to the manufacturer’s instructions. 

Real-Time PCR was performed on a CFX384 Touch Real-Time PCR Detecting System (Bio-Rad) and the data obtained were analyzed using the software provided. The reaction mixture comprised 2 μL cDNA template, 5 μL SYBR Green PCR master mix (Roche Biochemicals, IN), and 0.2 μL of 10 μM forward and reverse primers in a 10-μL reaction volume. The PCR protocol comprised one 20-s denaturation cycle at 95 °C, followed by 40 cycles of denaturation at 95 °C for 3 s, and an annealing/extension step at 60 °C for 30 s. 

We assessed changes in the mRNA levels of *CLOCK, BMAL1, PER 1–3, CRY 1–2, Rev-erb-α* (*Nr1d1*), *Rev-erb-β* (*Nr1d2*), and *D site of albumin promoter binding protein (DBP)* in leukocytes. The *TATA-box binding protein* (*TBP*) was used as the endogenous control. The primer sequences are shown in Appendix A. Clock gene expression levels were normalized to the TBP expression level at each time point and evaluated using a calibration curve method. 

### 2.4. Analysis

#### 2.4.1. Analysis of Circadian Rhythm

The time courses of the core body temperature and clock gene expression were evaluated using cosinor analysis; the cosinor technique was applied to each subject’s data using the least-squares regression method to estimate the phase of the circadian variation: time of maximum in the fitted cosine function, amplitude (half the difference between the minimum and maximum in the fitted cosine function), and mesor (rhythm-adjusted mean). 

#### 2.4.2. Statistical Analysis

All data are presented as means ± SEM. To detect a time shift in the circadian rhythms of the core body temperature and heart rate variability, the circular correlation coefficients between the 2 dietary conditions were evaluated. The circular correlation coefficient was calculated as the cross-correlation coefficient between the 2 time series on the circular time axis (clock time coordinate). The 06:00, 12:00, and 18:00 time-points were defined as the baseline for the post-meal gene expression, because clock gene expression levels reflect food-related metabolic changes, even in the postprandial state [35]. One-way analysis of variance (ANOVA) was used to evaluate the effect of a meal on the clock gene expression. As a post-hoc test, multiple comparisons using Dunnett’s test were conducted. Periodicity in the peak times of the cosinor curves was assessed by the Rayleigh test, which is an angle statistic and the most powerful invariant test for non-uniformity. Given the non-uniformity in the peak time of the cosinor curves, differences between the 2 dietary conditions were analyzed using a paired *t*-test. The differences in the amplitude and mesor were analyzed using a paired *t*-test. The level of significance was set at 5%, and *p* values less than 0.05 were considered statistically significant. The *p*-values from statistical tests were converted into false discovery rate *q*-values using the *Q*-value method, to correct Type I errors due to repeated statistics, and a false discovery rate *q*-value < 0.1 was considered statistically significant. All statistical analyses were performed using R version 3.6.0 (R Foundation for Statistical Computing, Vienna, Austria. http://www.R-project.org/). 

## 3. Results

Age, height, and body weight of the subjects were 25.1 ± 0.9 years, 173.4 ± 2.6 cm, 71.7 ± 4.2 kg, respectively. According to World Health Organization criteria [36], two subjects were overweight. Their chronotype was 58.1 ± 2.2, i.e., morning or intermediate type.

### 3.1. Core Body Temperature

Figure 2A shows double plots of the mean value of the core body temperature in the two dietary conditions. The core body temperature variations followed a 24-h cycle with a decrease in the night-time and an increase in the day-time. The amplitude of the cosinor analysis in the 3M and 2M conditions was 0.39 ± 0.02 and 0.42 ± 0.02 °C, respectively (*p* = 0.356). The mesor of the cosinor analysis was 36.71 ± 0.08 and 36.68 ± 0.06 °C, respectively (*p* = 0.732). The peak time of the cosinor analysis was 15.99 ± 0.28 h and 17.12 ± 0.34 h; the peak time of the raw data was 17.70 ± 0.68 h and 19.38 ± 0.45 h in the 3M and 2M conditions, respectively. The peak time differences between the cosinor analysis and raw data were 2.54 ± 0.24 h and 2.24 ± 0.35 h, respectively (Supplemental Appendix A). Thus, the peak time could not be estimated using cosinor analysis. Figure 3A shows the circular correlation of core body temperature in the two dietary conditions and the statistical significance of a non-zero mean (42.0 ± 16.2 min, *p* = 0.038).

### 3.2. Dim Light Melatonin Onset (DLMO)

The DLMO was 21:00–22:00 for the 3M condition and 20:00–22:30 for the 2M condition, although we were unable to determine the DLMO in 4 of 20 cases, as the level of melatonin did not continue to rise. The DLMO did not differ significantly between the two dietary conditions (Figure 4).

### 3.3. Heart Rate Variability

Figure 2B–D shows double plots of the mean value for heart rate, LF/HF, and log HF in the two dietary conditions. Heart rate was higher during the day than at night in both dietary conditions, particularly when subjects consumed a meal. LF/HF was higher during the day than at night in both dietary conditions, while log HF increased during the night and decreased during the day. Figure 3B–D shows the circular correlation coefficients of heart rate, LF/HF, and log HF in the two dietary conditions, and no statistical significance of a non-zero mean. 

### 3.4. Clock Genes

The expression level of *PER2* was higher at 09:00 in the 3M condition (*p* = 0.041, *q* = 0.054), and expression levels of *CRY1* and *CRY2* were higher at 12:00 in the 2M condition (*p* = 0.026, *q* = 0.041 and *p* = 0.016, *q* = 0.030, respectively; Figure 5A). The expression levels of *PER1* and *PER3* were lower at 15:00 for both dietary conditions (3M: *p* = 0.005, *q* = 0.014 and *p* = 0.017, *q* = 0.031, respectively, 2M: *p* = 0.003, *q* = 0.011 and *p* = 0.007, *q* = 0.017, respectively) and the expression levels of *PER1*, *PER2*, *PER3*, *CRY1*, *CRY2*, and *NR1D2* were lower at 18:00 in the 2M condition (*p* < 0.001, *q* = 0.003, *p* < 0.001, *q* = 0.004, *p* = 0.001, *q* = 0.003, *p* = 0.010, *q* = 0.019, *p* = 0.045, *q* = 0.059, and *p* = 0.004, *q* = 0.012, respectively; Figure 5B). The expression level of *NR1D1* was higher at 23:00 for both dietary conditions (3M: *p* = 0.016, *q* = 0.030; 2M: *p* < 0.001, *q* < 0.001), and in the 2M condition, the expression level of *NR1D1* was higher at 21:00 (*p* = 0.010, *q* = 0.019) and the expression of *NR1D2* and *DBP* were higher at 23:00 (*p* < 0.001, *q* = 0.001 and *p* = 0.001, *q* = 0.005, respectively; Appendix A). The expression levels of the other clock genes did not differ significantly pre- and post-meal. 

The time-course of the expression of each clock gene is shown as a z-score. Figure 6A–J shows the mean expression level of the clock genes in the two dietary conditions. Figure 7A–J shows the peak expression time of these clock genes and the results of the Rayleigh test in the two dietary conditions. The results of the Rayleigh test indicated a significant difference in the peak time of expression of *PER1* (3M: *p* = 0.003, 2M: *p* < 0.001), *PER2* (3M: *p* = 0.008, 2M: *p* < 0.001), *PER3* (3M: *p* = 0.011, 2M: *p* < 0.001), and *NR1D1* (3M: *p* = 0.003, 2M: *p* = 0.001) in the two dietary conditions, but the peak expression times of the other genes (*CLOCK*, *BMAL1*, *CRY1*, *CRY2*, *NR1D2*, and *DBP*) did not differ significantly. The peak expression time of *PER1*, *PER2*, *PER3*, and *NR1D1* did not differ significantly between the two dietary conditions (*p* = 0.151, *p* = 0.506, *p* = 0.987, and *p* = 0.178, respectively). The amplitude and mesor of these clock genes (*CLOCK*, *BMAL1*, *PER1*, *PER2*, *PER3*, *CRY1*, *CRY2*, *NR1D1*, *NR1D2*, and *DBP*) also did not differ significantly between the two dietary conditions (Table 1).

## 4. Discussion

In this study, the effects of skipping breakfast over the short-term (6 consecutive days) on the phase of the core body temperature and clock gene expression in healthy young individuals were evaluated. On the sixth consecutive day of skipping breakfast, the diurnal rhythm of the core body temperature exhibited a phase delay. On the contrary, circadian rhythms of DLMO, heart rate variability, and expression of the clock genes (*CLOCK*, *BMAL1*, *PER1*, *PER2*, *PER3*, *CRY1*, *CRY2*, *NR1D1*, *NR1D2*, and *DBP*) were not affected by skipping breakfast. The first meal of the day was delayed by 5.5 h in the breakfast-skipping condition, and the phase of a biologic clock marker, i.e., the core body temperature, was delayed by 1 h. 

Circadian clocks organize behavior and physiology to adapt to daily environmental cycles. Many studies report that chronic misalignment between the endogenous circadian rhythm and environmental/social rhythms is a significant risk factor for various disorders, including sleep disorders, metabolic syndrome, cardiovascular diseases, and cancer [27,37,38]. In particular, breakfast consumption may be favorably associated with dietary quality, body composition, and markers of chronic disease risk [39,40]. Most of the previous human studies [41,42,43,44] were observational and little is known about the effects of meal timing per se on human circadian rhythms; thus, there is currently no physiologic evidence to explain these effects in humans. Mealtimes and number of meals consumed differ greatly from culture to culture and through time. Indeed, the timing of food intake is a modifiable behavior that may influence energy metabolism, i.e., even when the energy intake is the same each day, eating during the phase of inactivity leads to weight gain, and is consequently a risk for obesity [45].

The effect of skipping breakfast on the human core body temperature has not been sufficiently studied. Previous studies measured sublingual temperature, but it was not measured over a long period of time to assess the amplitude and phase of the body temperature rhythm [13,14]. In the present study, the circadian rhythm of the core body temperature exhibited a phase delay on the sixth consecutive day of skipping breakfast despite the conditions of the sleep–wake cycle being equal in both conditions. Although the mean value of the core body temperature was reduced in the morning when breakfast was skipped, the amplitude and mesor of the cosinor analysis were not affected by skipping breakfast. The present result showing a reduction in the core body temperature is consistent with findings from previous animal studies. The core body temperature is reduced in the nocturnal period by nocturnal fasting for 4 days [18] or 2 weeks [19,46]. Moreover, the degree of the core body temperature reduction gradually increases and then stabilizes after 4 days of skipping breakfast [46]. Although light is a powerful external cue to synchronize the organism’s biologic rhythms to the earth’s 24-h light/dark cycle [17], meal timing is also an important factor. Significant differences were found in the core body temperature rhythm despite the fact that the light–dark cycle was equal in both conditions. Contrary to our expectations, the phase of the core body temperature was delayed by only approximately 1 h, even though in the present study the first meal in the breakfast skipping condition was 5.5 h later than that in the three-meal condition. A possible explanation for the smaller effect of breakfast skipping on the circadian rhythm of the core body temperature is that the light/dark cycles were common between both meal conditions. In addition, the rhythm of the core body temperature might have been masked by the thermic effects of the meal; the lack of a thermic effect of breakfast, and a higher thermic effect of lunch and dinner due to the larger meal size in the breakfast-skipping condition. Further studies are needed to investigate the thermic effects of the meal on circadian rhythms.

In the present study, skipping breakfast did not affect saliva DLMO, which is consistent with recent findings reported by Wehrens et al. [21]. They adopted a 5-h delay in mealtimes, but detected no change in markers of the SCN clock (melatonin and cortisol). This finding suggests that mealtime, i.e., skipping breakfast and shifting the timing of the three meals, would not alter the DLMO, a reliable marker of the SCN clock [47,48]. 

Heart rate variability can be used as a measure of the activity of both components of the autonomic nervous system. Yoshizaki et al. [49], recruited young male subjects with a habit of regularly skipping breakfast (meals at 13:00, 18:00, and 23:00; control group) and participants were asked to eat earlier (meals at 08:00, 13:00, and 18:00) for 2 weeks; early mealtime group). They reported that meal consumption earlier in the day resulted in a phase shift in heart rate variability in a 24-h period rhythm, and significant phase advances in LF power (−3.2 ± 1.2 h) and the ratio of HF power to total power (%HF, −1.2 ± 0.5 h, vagal nervous system activity) in the early mealtime group as compared with the control group after 2 weeks, using a double cosinor analysis. In our heart rate variability analysis, no cosinor techniques were used, because the multiple-peaked circadian patterns of the 24-h heart rate variability were not well-approximated by cosinor-type curves. Therefore, to detect a time shift in the heart-rate-variability circadian pattern, circular correlation coefficients between the two dietary conditions were evaluated. Our results indicated that 6 consecutive days of skipping breakfast had no effect on the phase of heart rate variability, differing from findings of Yoshizaki et al. [49]. Possible reasons for the apparent discrepancy between the present findings and the findings reported by Yoshizaki et al. [49] are: (1) meal timing differed at only one time-point (with or without breakfast) in the present study, whereas the timing of all meals was delayed in the previous study; (2) the double cosinor analysis could not be fitted in the present study, because the circadian patterns of the 24-h heart rate variability showed approximately three dominant peaks corresponding to the mealtimes, whereas it could be fitted in the previous study; and (3) the meal intervention period was 6 days in the present study and 2 weeks in the previous study.

In the present study, we evaluated diurnal changes in clock gene expression by sampling leukocytes eight times in a 24-h period because repeated blood sampling is feasible, and the clock genes in leukocytes exhibit rhythmic expression [50,51,52]. Moreover, gene expression in leukocytes may provide an accessible window to the multiorgan transcriptome [53,54]. Jakubowicz et al. conducted a human study to assess the effect of a single incident of skipping breakfast on clock gene expression in white blood cells measured at 8:30, 12:00, and 15:30 [55]. Compared with the value before breakfast (8:30), the expression level of *PER1* and *CRY1* was lower but that of *CLOCK* was higher after breakfast (at 12:00). Even without breakfast, the expression level of *BMAL1* and *CRY1* was upregulated during this time period. Although their study suggested that the expression of some, but not all, clock genes was affected by breakfast, the clock gene expression between the two dietary groups was not compared. In the present study, the expression level of the clock genes in both the positive feedback loop (*CLOCK* and *BMAL1*) and the negative feedback loop (*PER1–3* and *CRY1–2*) at 9:00 and/or 12:00 was higher than that at 6:00 regardless of breakfast intake. Regarding the postprandial response, the results may be different for the following reasons: (1) the reference time was different, i.e., it had been delayed 150 min (8:30) in the previous study; (2) the quantity and content of the meals differed, i.e., low energy intake and content of protein was high (carbohydrate was low) in the previous study; and (3) the sampling time was different, i.e., two points (0 and 210 min or 210 and 420 min) in the previous study. 

Several reports suggest that meal timing exerts a pivotal influence on peripheral clocks and clock output systems involved in the regulation of metabolic pathways. Previous studies reported that skipping breakfast for 4 days leads to a phase delay in the expression of *BMAL1* in mouse liver and fat tissue [18], skipping breakfast for 2 weeks leads to phase delays in the expression of *BMAL1*, *CLOCK*, *PER2*, and *CRY1* in rat hepatic tissue [19], and skipping breakfast for 4 weeks leads to phase delays in the expression of *BMAL1*, *CRY1*, *NR1D1*, and *PER1~3* in rat hepatic tissue [56]. A previous human study [21] reported that compared with a regular meal pattern (meals at 07:00, 12:00, and 17:00), delayed meals (meals at 12:00, 17:00, and 22:00) induce a significant phase delay in *PER2* mRNA rhythms in white adipose tissue by 0.97 ± 0.29 h. An effect of skipping breakfast on the peak time of clock gene expression was not observed in the present study, although Kajimoto et al. suggested that feeding-induced insulin release resets peripheral circadian clocks in humans [57]. Possible explanations for the lack of an effect of skipping breakfast on clock gene expression include: (1) the expression of clock genes was evaluated in leukocytes, not hepatic tissue [18,19,56] or white adipose tissue [21]; (2) there were large intra-individual differences, i.e., the Rayleigh test suggested non-uniformity of the acrophase in only 4 of 10 clock genes; and (3) it was a relatively short-term dietary intervention, i.e., 6 days. Further studies are needed to investigate the chronic effects of skipping breakfast.

The core body temperature is increased even without eating breakfast, but it was greatly increased with breakfast. Young people who tend to be night type could prevent the delay of their core body temperature by having breakfast without changing their bedtime. It may have been possible to partially prove the importance of eating breakfast.

### Limitations 

This study has several limitations. First, some tissue-specific responses of peripheral tissue clocks have been demonstrated [21]. To generalize findings concerning the effects of skipping breakfast on the expression of clock genes in the present study, the effects of skipping breakfast on clock gene expression in other peripheral tissues should be evaluated. Second, the sample size was small, and the age of the subjects was limited. To generalize the present findings, experiments with young healthy women, middle-aged, and elderly are warranted. Third, the beginning of exposure to dim light might have been a bit late to assess DLMO, because we could not clearly show DLMO in 4 of 20 cases in the present study. Forth, habitual breakfast eaters were recruited for the present study, but there was some ambiguity in the inclusion criteria for breakfast eaters as subjects in the study. Lastly, we measured core body temperature, DLMO, heart rate variability, and expression of clock genes only on the sixth day of skipping breakfast, so we could not evaluate the time-course of the changes.

## 5. Conclusions

In conclusion, in healthy young humans, a 6-day breakfast-skipping intervention did not affect the 24-h heart rate variability, clock gene expression, or DLMO, but did cause a phase delay in the core body temperature in the 2M condition compared with the 3M condition, even though the sleep–wake cycle remained unchanged. Chronic effects of breakfast skipping on circadian rhythms remain to be studied.

## Figures and Tables

**Figure 1 nutrients-12-02797-f001:**
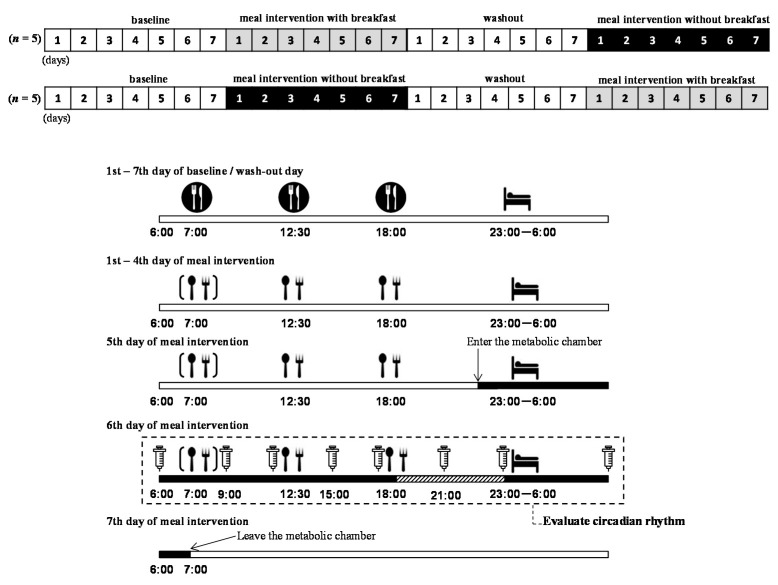
Study protocol.

**Figure 2 nutrients-12-02797-f002:**
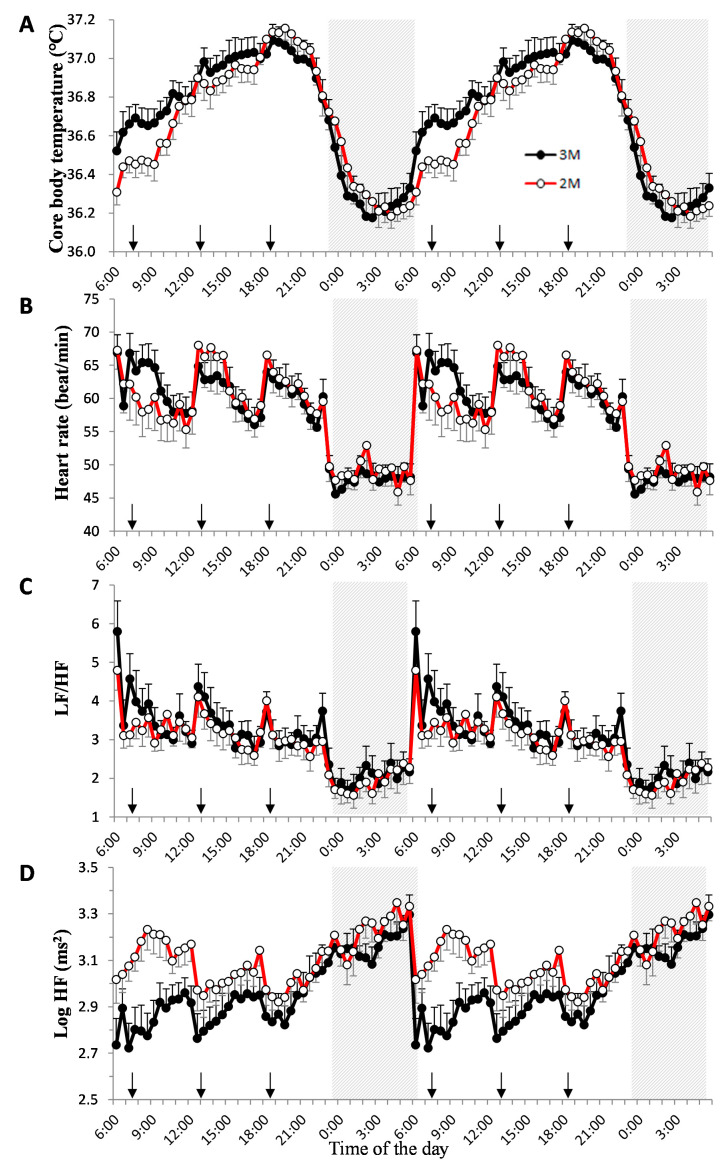
Diurnal variations in core body temperature, heart rate, LF/HF, and log HF. Double plots of the mean value ± SEM for core body temperature (**A**), heart rate (**B**), LF/HF (**C**), and log HF (**D**) in the 2 dietary conditions. Sleep hours (23:00–06:00) are indicated in the gray zone. The closed circles indicate the 3-meal per day diet (3M) condition (
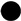
) and the open circles indicate the 2-meal per day diet (2M) condition (
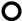
). Arrows indicate the meal timing. LF/HF, the power ratio of low frequency to high frequency; HF: high frequency.

**Figure 3 nutrients-12-02797-f003:**
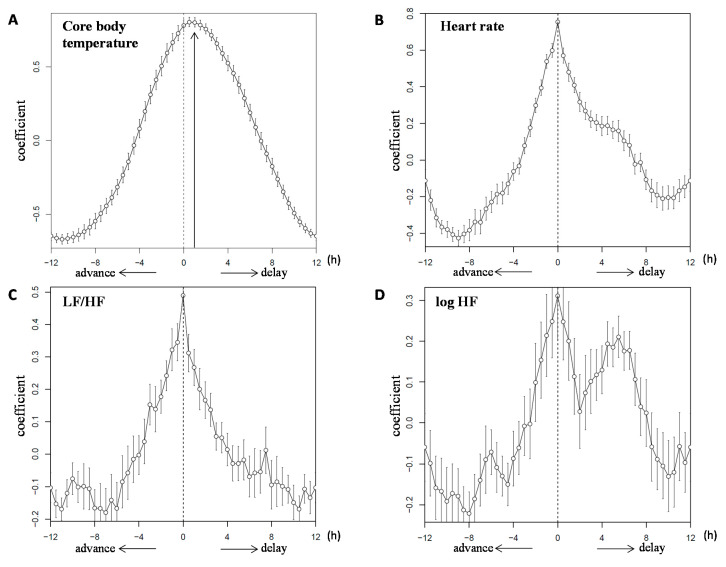
Circular correlation of core body temperature, heart rate, LF/HF, and log HF. The circular correlation of the mean value ± SEM for core body temperature (**A**), heart rate (**B**), LF/HF (**C**), and log HF (**D**) in the 2 dietary conditions. Note that the 2M condition was plotted according to the 3M condition. LF/HF, the power ratio of low frequency to high frequency; HF: high frequency.

**Figure 4 nutrients-12-02797-f004:**
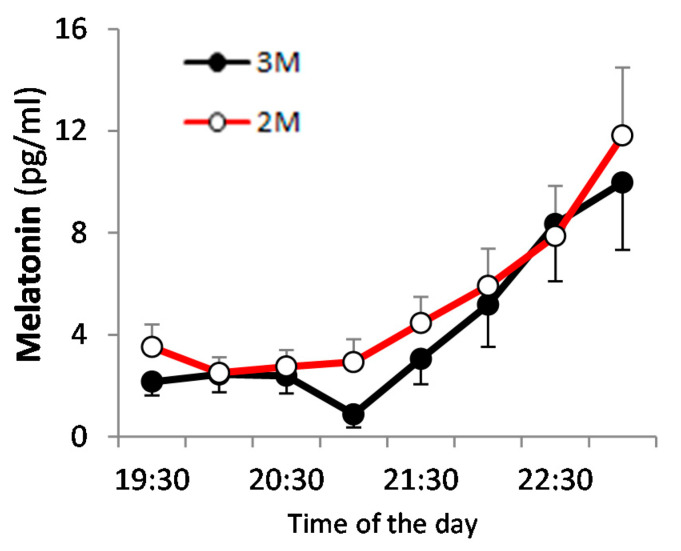
Dim light melatonin onset (DLMO) in saliva. Saliva melatonin profiles are presented as the mean value ± SEM for the 2 dietary conditions. The closed circles indicate the 3M condition (
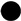
) and the open circles indicate the 2M condition (
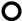
).

**Figure 5 nutrients-12-02797-f005:**
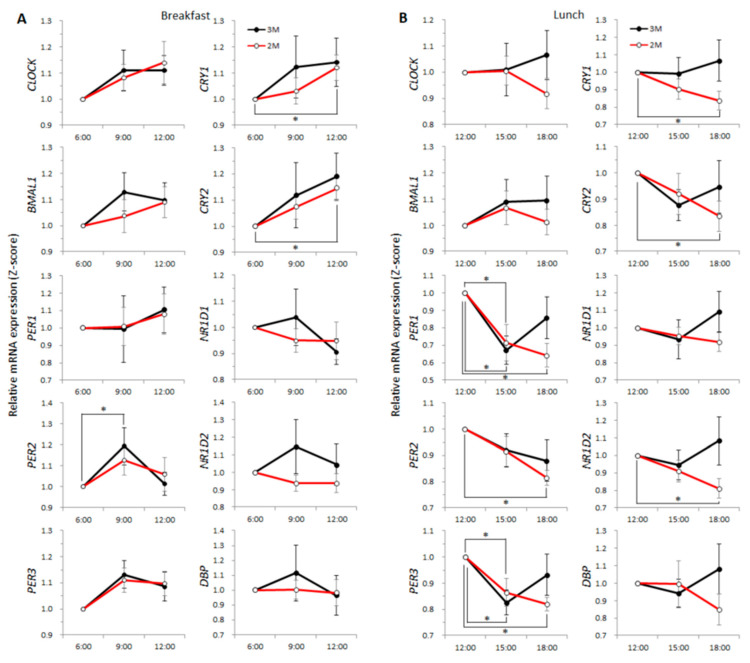
Gene expression with or without breakfast and after lunch. The clock gene expression profiles are presented as the mean value ± SEM for the 2 dietary conditions. Left panels (**A**) show with or without breakfast; blood samples were collected after overnight fasting (time-point 6:00), 2 h after breakfast (3M) or fasting continued (2M) (9:00), and 4 h after breakfast or fasting continued (12:00). Right panels (**B**) show the profiles after lunch; blood samples were collected 5 h after breakfast (3M) or no breakfast (2M) (time-point 12:00), 2.5 h after lunch (15:00), and 5.5 h after lunch (18:00). The closed circles indicate the 3M condition (
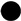
) and the open circles indicate the 2M condition (
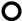
). One-way ANOVA was used to evaluate the effect of a meal on the expression of each clock gene. As a post-hoc test, multiple comparisons using Dunnett’s test were conducted.

**Figure 6 nutrients-12-02797-f006:**
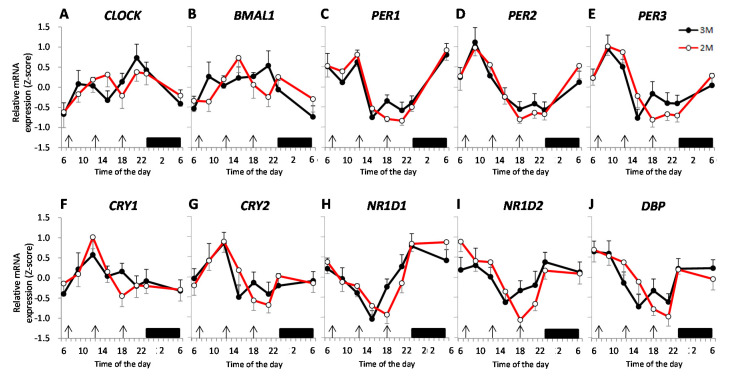
Expression profiles for circadian clock genes in leukocytes. Data are presented as the mean standardized z-score value ± SEM for relative expression of *circadian locomotor output kaput* (*CLOCK*) (**A**), *brain and muscle ARNT-like protein 1* (*BMAL1*) (**B**), *period* (*PER1*) (**C**), *PER2* (**D**), *PER3* (**E**), *cryptochrome* (*CRY*)1 (**F**), *CRY2* (**G**), *Rev-erb-α* (*NR1D1*) (**H**), *Rev-erb-β* (*NR1D2*) (**I**), and *D site of albumin promoter binding protein* (*DBP*) (**J**) in the 2 dietary conditions. The closed circles indicate the 3M condition (
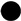
) and the open circles indicate the 2M condition (
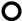
). Arrows indicate the meal timing. Sleep hours (23:00–06:00) are indicated by the black zone.

**Figure 7 nutrients-12-02797-f007:**
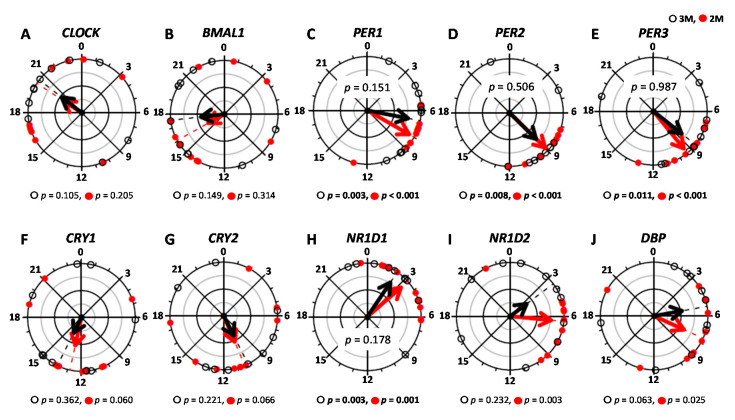
The peak time of clock genes. The peak times were evaluated using cosinor analysis and are presented: *CLOCK* (**A**), *BMAL1* (**B**), *PER1* (**C**), *PER2* (**D**), *PER3* (**E**), *CRY1* (**F**), *CRY2* (**G**), *NR1D1* (**H**), *NR1D2* (**I**), and *DBP* (**J**) in the 2 dietary conditions. The open circles indicate the 3M condition (
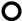
) and the closed circles indicate the 2M condition (
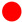
).The black arrow indicates the distribution width of the 3M condition (
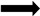
) and the red arrow indicates the distribution width of the 2M condition (
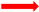
). The *p*-value under the pie chart is the result of the Rayleigh test. The *p*-value in the pie chart was the result of the paired *t*-test. Radius of the pie chart = 1.

**Table 1 nutrients-12-02797-t001:** Comparison of clock genes by cosinor fitting ^1.^

Parameter	Peak Time (h)				Amplitude				Mesor			
	3M	2M	*p*-Value ^2^	*q*-Value ^3^	3M	2M	*p*-Value ^2^	*q*-Value ^3^	3M	2M	*p*-Value ^2^	*q*-Value ^3^
*CLOCK*	18.51 ± 1.70	19.73 ± 1.62	−		0.11 ± 0.02	0.11 ± 0.02	0.820	0.417	0.90 ± 0.10	0.98 ± 0.03	0.400	0.259
*BMAL1*	16.60 ± 1.59	17.44 ± 1.83	−		0.11 ± 0.02	0.10 ± 0.02	0.360	0.248	0.99 ± 0.09	0.98 ± 0.08	0.975	0.467
*PER1*	6.68 ± 0.90	8.10 ± 0.62	0.151	0.144	0.31 ± 0.11	0.23 ± 0.05	0.433	0.273	1.01 ± 0.25	0.81 ± 0.15	0.187	0.169
*PER2*	10.44 ± 1.49	8.99 ± 0.48	0.506	0.297	0.19 ± 0.03	0.15 ± 0.02	0.238	0.193	1.10 ± 0.13	1.08 ± 0.10	0.418	0.268
*PER3*	9.40 ± 1.28	9.42 ± 0.60	0.987	0.456	0.15 ± 0.03	0.14 ± 0.02	0.737	0.386	1.17 ± 0.19	1.23 ± 0.19	0.456	0.284
*CRY1*	12.90 ± 2.04	13.05 ± 1.45	−		0.20 ± 0.03	0.13 ± 0.02	0.081	0.093	1.22 ± 0.14	1.25 ± 0.09	0.725	0.385
*CRY2*	12.61 ± 1.83	10.09 ± 1.43	−		0.13 ± 0.03	0.11 ± 0.01	0.617	0.353	1.12 ± 0.10	1.19 ± 0.10	0.350	0.245
*NR1D1*	7.29 ± 2.70	5.65 ± 2.09	0.178	0.163	0.21 ± 0.03	0.18 ± 0.03	0.518	0.308	0.14 ± 0.07	0.08 ± 0.04	0.857	0.430
*NR1D2*	9.81 ± 2.50	8.34 ± 1.63	−		0.16 ± 0.02	0.15 ± 0.01	0.713	0.383	1.13 ± 0.07	1.20 ± 0.09	0.515	0.308
*DBP*	6.79 ± 1.56	8.98 ± 1.48	−		0.23 ± 0.03	0.21 ± 0.03	0.647	0.364	1.21 ± 0.14	1.28 ± 0.15	0.548	0.322

^1^ Values are means ± SEM; ^2^
*P* values for peak times, amplitude, and mesor were analyzed by paired *t*-test; ^3^ The *q*-values were analyzed using the *Q*-value method; 3M; with breakfast, 2M; without breakfast.

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
