# Peer review of "Skipping Breakfast for 6 Days Delayed the Circadian Rhythm of the Body Temperature without Altering Clock Gene Expression in Human Leukocytes"

_nutrients, 2020, doi:10.3390/nu12092797_

Round 1
Reviewer 1 Report
The paper entitled “Skipping breakfast for 6 days delayed the circadian rhythm of the body temperature but did not alter the peak time of clock gene expression in human leukocytes” by Ogata H et al. aims to investigate the effects of skipping breakfast on core body temperature, dim light melatonin onset, heart rate variability, and clock gene expression. The topic is of a certain interest and novelty, the methods are clearly explained, the results are interesting and conclusions sound. Figure 1 is very informative.
Minore remarks:
- The title is too long.
- In the inclusion/exclusion criteria, what about usual physical activity?
- Why were overweight subjects also enrolled?
- Was the energy expenditure considered? And, if so, how?
- In the discussion, some points about the practical and clinical implications of your findings should be added.
Author Response
- The title is too long.
Response: We changed the title as follows; Skipping breakfast for 6 days delayed the circadian rhythm of the body temperature without altering clock gene expression in human leukocytes.
- In the inclusion/exclusion criteria, what about usual physical activity?
Response: The subjects were instructed to refrain extreme exercise during the meal intervention, because we focused on the meal timing. But in the exclusion criteria, usual physical activity was not considered.
- Why were overweight subjects also enrolled?
Response: Exclusion criteria did not contain overweight. Because in this experiment, the amount of energy intake at one time during the breakfast skipping condition was large, so we prioritized the collection of the subject who could eat the provided meal. Note that body mass index of two subjects was 25.2 and 29.3 kg/m2.
- Was the energy expenditure considered? And, if so, how?
Response: Yes, it was. We measured energy expenditure using whole-body indirect calorimetry (Please see Lines 105–106 for reference [29]).
- In the discussion, some points about the practical and clinical implications of your findings should be added.
Response: Thank you for your very insightful comment. We added the manuscript (Lines 443–446).
Reviewer 2 Report
Chrono nutrition is an emerging area of nutritional research. This paper details secondary results from a study investigating energy expenditure and substrate oxidation (previously published). The results add to limited knowledge in this area.
Abstract:
Line 26: "breakfast is the most important meal of the day' - should be rephrased similar to the introduction ' often described as..'
Study protocol:
- Sample size (lines 95-98) – please state the primary outcome of the preliminary study that this study sample size was derived from.
- Details of the meals provided including macronutrient profile should be included in the supplementary material. Nutrient profiles have also been shown to impact HRV measurements.
- Were any sleep outcomes assessed? E.g. latency, quality, efficiency? – did these differ between the conditions?
- Were order effects tested?
Results:
- Figure 2 – LF / HF, all abbreviations to be written in full in the figure legend.
- Line 254: 3.2 DLMO – write in full as subheading
- Figures 5, 6 and 7 – colour contrast rather than grey scale would improve readability of these figures.
Limitations:
- Sample size – is the sample size used adequate to detect changes in the outcome measures of this study? Need to relate this to the strength / application of the results.
- What would the value of baseline DLMO measurement be to future studies?
Author Response
Abstract:
Line 26: "breakfast is the most important meal of the day' - should be rephrased similar to the introduction ' often described as..'
Response: We have changed the manuscript (Line 26).
Study protocol:
- Sample size (lines 95-98) – please state the primary outcome of the preliminary study that this study sample size was derived from.
Response: According to our previous study (Obes Res Clin Pract 2014;8(3):e249–57), a power analysis revealed that a sample size of 8 subjects is required to provide 80% power to detect a 5% difference between 2 dietary conditions in the 24-h mean blood glucose. In this document, the blood glucose level is not treated as data, so it is not included in the main text.
- Details of the meals provided including macronutrient profile should be included in the supplementary material. Nutrient profiles have also been shown to impact HRV measurements.
Response: Thank you for your very insightful comment. We added the manuscript (Line 121), because of one sentence.
- Were any sleep outcomes assessed? E.g. latency, quality, efficiency? – did these differ between the conditions?
Response: Unfortunately, we did not assess sleep outcomes. In the future, comparisons between the conditions should be considered, as you pointed out.
- Were order effects tested?
Response: Yes, they were. In order to prevent the order effects, we conducted the experiment with a protocol that considers counterbalanced.
Results:
- Figure 2 – LF / HF, all abbreviations to be written in full in the figure legend.
Response: We have added the manuscript (Lines 248–249, 254).
- Line 254: 3.2 DLMO – write in full as subheading
Response: We have changed the manuscript (Line 255).
- Figures 5, 6 and 7 – colour contrast rather than grey scale would improve readability of these figures.
Response: Thank you for your very insightful comment. We changed color contrast (Figures 2, 4, 5, 6, and 7).
Limitations:
- Sample size – is the sample size used adequate to detect changes in the outcome measures of this study? Need to relate this to the strength / application of the results.
Response: We added a small sample size as a study limitation (Lines 450–452).
- What would the value of baseline DLMO measurement be to future studies?
Response: We believe that DLMO measurement will be useful for future human experiments because the index is considered a gold-standard proxy for the SCN phase and it can be obtained by a simple measurement method.